# Proteomics and Microbiota Conjoint Analysis in the Nasal Mucus: Revelation of Differences in Immunological Function in *Manis javanica* and *Manis pentadactyla*

**DOI:** 10.3390/ani14182683

**Published:** 2024-09-14

**Authors:** Qing Han, Yepin Yu, Hongbin Sun, Xiujuan Zhang, Ping Liu, Jianfeng Deng, Xinyuan Hu, Jinping Chen

**Affiliations:** 1Guangdong Key Laboratory of Animal Conservation and Resource Utilization, Guangdong Public Laboratory of Wild Animal Conservation and Utilization, Institute of Zoology, Guangdong Academy of Sciences, Guangzhou 510260, China; hanq920920@163.com (Q.H.); yepin@giz.gd.cn (Y.Y.); zhangxj67@giz.gd.cn (X.Z.); pingliu0330@126.com (P.L.); 2Shenzhen Natural Reserve Management Center, Shenzhen 518115, China; sun_hongbin@126.com (H.S.); xiaohouniao57@foxmail.com (J.D.); pipih01@foxmail.com (X.H.)

**Keywords:** *Manis javanica*, *Manis pentadactyla*, nasal mucosa microbiota, TMT-proteomics

## Abstract

**Simple Summary:**

Pangolins, the only mammals covered in scales, play a crucial role in forest ecosystems as they specialize in myrmecophagy. Unfortunately, all eight pangolin species are critically endangered and susceptible to various pathogenic microorganisms, causing mass mortality, especially in captive *Manis pentadactyla*. However, information regarding the function of the immune system is lacking, which limits the development of effective rescue methods and subsequently hinders population rejuvenation. This study aimed to investigate the differences in the immunity of *Manis javanica* and *Manis pentadactyla* through proteomics and microbiotas conjoint analysis. Our findings revealed that *Manis pentadactyla* owned more pathogenic bacteria and neutralized through a powerful transferrin system. *Manis javanica* possessed stronger anti-inflammatory ability, which might be due to the structural deficiency of C5a. This study elucidates the distinct immune factors and microbiomes in *Manis javanica* compared to *Manis pentadactyla*, offering a foundational understanding for future immunotherapy research.

**Abstract:**

All eight pangolin species, especially captive *Manis pentadactyla*, are critically endangered and susceptible to various pathogenic microorganisms, causing mass mortality. They are involved in the complement system, iron transport system, and inflammatory factors. *M. pentadactyla* exhibited a higher abundance of opportunistic pathogens, *Moraxella*, which potentially evaded complement-mediated immune response by reducing C5 levels and counteracting detrimental effects through transferrin neutralization. In addition, we found that the major structure of C5a, an important inflammatory factor, was lacking in *M. javanica*. In brief, this study revealed the differences in immune factors and microbiome between *M. javanica* and *M. pentadactyla*, thus providing a theoretical basis for subsequent immunotherapy.

## 1. Introduction

*Manis javanica* and *M. pentadactyla* are the main pangolins in China, and classified as critically endangered following the International Union for Conservation of Nature (IUCN) Red List and Appendix I (Available online at: https://cites.org/eng/app/appendices.php#, accessed on 22 March 2023) of the Convention on International Trade in Endangered Species of Wild Fauna and Flora (CITES I) due to the scarcity of quantity [1]. Pangolin specializes in myrmecophagy, that plays a crucial role in protecting the forests from the attack of ants and termites, thereby maintaining the delicate balance of the ecosystem [2]. Numerous studies have demonstrated the susceptibility of pangolin to many pathogenic microorganisms causing serious surface or visceral inflammation [3,4,5]. However, the information regarding the function of the immune system is lacking, severely limiting the rescue work for the population rejuvenation of pangolins. In addition, we found that the inflammatory response of *M. pentadactyla* was more difficult to cure than that of *M. javanica*, leading to a higher mortality rate. Hence, analyzing the differences in the immune systems of *M. javanica* and *M. pentadactyla* is essential to help in pangolin rescue and population restoration.

The mucosal immune system is an important place for organisms to exert immune function, including physical (mucus layer, epithelial cells, etc.), chemical (proteolytic enzymes, anti-infective effecter proteins, etc.), and biological (immune cells, immunoglobulins, etc.) barriers. They work together to help innate immunity and adaptive immunity to resist the invasion of pathogenic microorganisms [6]. In addition, there is a large number of symbiotic bacteria in the mucosal immune system, maintaining a dynamic balance. They participate in innate immune and promoting goblet cells to secrete mucus and are involved in anti-inflammatory responses through TLR or other pathways [7]. In addition, the bacteria in the mucosal immune system also participate in the response and regulation of adaptive immunity, inducing the differentiation and response of T cell subsets, such as Th1, Th17, and Treg [8]. Mucosal immune system keeps a low response state to symbiotic bacteria and covers pathogens with immunoglobulin (sIgA) to monitor and prevent their invasion of mucosal epithelial cells [9]. A previous study discovered that some molecules of innate immunity existed in the skin of *M. javanica*, confirming pangolin possessed the basis of the mucosal immune system, whereas the deeper research was lacking [10].

Proteomics can reveal biochemical processes and mechanisms at the protein level [11]. Based on high-resolution mass spectrometry, tandem mass tag (TMT) quantifies the proteins from 16 different samples concurrently. It has been applied to explain the physiological and pathological phenomena by testing mucus, such as endocervical mucus [12], nasal mucus [13], and skin mucus [14], from humans and animals. In addition, with the application of genomics in *M. javanica* and *M. pentadactyla*, multitudinous molecules related to the immune system were annotated, which provided a foundation for the study of pangolin proteomics [15]. Therefore, this study was performed to clarify the differences in the immune level between *M. javanica* and *M. pentadactyla*. First, the nasal mucus was collected without damage, acting as a vital component in mucosal immunity. Then, TMT analysis combined with liquid chromatography–tandem mass spectrometry (LC-MS/MS) was performed to examine the proteomic changes. Bioinformatics prediction, recombinant protein verification, and nasal mucosa microbiome association analysis were performed integrally to deepen our understanding of the functional differences between pangolins, aiming to search for an effective strategy for rescue work.

## 2. Materials and Methods

### 2.1. Pangolin Sample Collection and Preparation

The nasal mucus of 10 pangolins (weight 5.71 kg ± 0.96 kg) was collected from the Wildlife Rescue Center of Shenzhen (Shenzhen City) and Guangdong Province (Guangzhou City); detailed information about their names, species, genders, and sampling locations is provided in Appendix A. The pangolins from two different Wildlife Rescue Centers were all in a healthy condition, although their feeding diets were not completely consistent. The mucus was collected into Eppendorf tubes, vigorously shaken, and centrifuged sequentially at 800× *g* for 10 min to remove large particles and cell debris (in the sediment). After centrifugation, the supernatant was further centrifuged at 14,000× *g* for 20 min, and the resulting supernatant was collected as the nasal mucus. The sediment comprised bacterial pellets from the nasal mucus. All samples were immediately stored at −80 °C until use.

### 2.2. TMT Labeling and LC-MS/MS Analysis

Next, 100 µg of protein sample was taken from nasal mucus, denatured, and alkylated with 5 mM DTT (Sigma-Aldrich, Shanghai, China) and 10 mM iodoacetamide (Sigma-Aldrich). The peptides were collected after freeze-drying and labeled with 126, 127, 128, 129, or 130 TMT reagents (Appendix A) using a TMT 6-Plex kit (Thermo Fisher Scientific, Waltham, MA, USA) following the manufacturer’s protocol. The TMT-labeled peptide mixture was resuspended and separated using the nanoflow liquid chromatography on Easy-nLC1000 (Thermo Fisher Scientific), and then injected into Q-ExactiveMS (Thermo Fisher Scientific) to perform LC-MS/MS analysis.

### 2.3. Protein Database Search and Bioinformatics Analysis

The raw data from LC−MS/MS were analyzed using Proteome Discoverer 2.3 (Thermo Fischer Scientific, Bremen, Germany) workflow. The data interpretation and protein identification were carried out using the latest *Manis* protein databases (downloaded from NCBI, 4 March 2023 version), configured with Proteome Discoverer 2.4 software for searching the datasets. The search parameters include 15 ppm and 0.02 Da mass tolerances for MS and MS/MS, respectively. Trypsin digestion allows two missed cleavages with fixed modifications of cysteine carbamidomethylation, N-terminal TMT10plex, and lysine TMT10plex; and with Acetyl and oxidation (M) as variable modifications. The value of max missed cleavages was 2. Additionally, the peptides were extracted using high peptide confidence. The calculation of a 1% false discovery rate (FDR) was used for searching the peptide sequence. The peptide sequences were searched in the database of *M. javanica* and *M. pentadactyla*, and a protein must have at least one unique peptide match with TMT ratios. The zero values were preserved over half of the samples containing valuable proteins, followed by the utilization of the Perseus algorithm (stochastic compensation) to compensate for the missing data. The data after compensation were performed by common protein normalization calculation. The principal component analysis (PCA), Venn diagram, volcano plots, and hierarchical clustering were achieved using one-way analysis of variance, *t* test, and R software version 3.6.1. The GO enrichment (http://www.geneontology.org/, accessed on 24 January 2022) analysis for identifying proteins was performed and classified into three categories: biological processes, cellular compartments, and molecular functions. The KEGG pathway enrichment (http://www.genemo.jp/kegg/, accessed on 24 January 2022) analysis was performed using the R function for functional pathway analysis. GO and KEGG with the corrected *p* value < 0.05, Fold change > 1.2 were considered significant in *t* test, and performed by Fisher’s exact test. The raw MS data were available via ProteomeXchange with identifier PXD054168.

### 2.4. DNA Extraction and Data Analysis of Microbiome

For microbiome analysis, the genomic DNA of bacteria pellet was extracted using a TaKaRa MiniBEST Bacteria Genomic DNA Extraction Kit (TaKaRa, Dalian, China) following the manufacturer’s protocols. The full-length 16S rRNA gene was amplified with 2× Taq Master Mix (Vazyme, Nanjing, China), followed by primers listed in Appendix A. Sequencing on an Illumina HiSeq platform (250 bp paired-end reads) was performed by Biomarker Technologies (Beijing, China). The original data have been uploaded to SRA data in National Center for Biotechnology Information (NCBI) database (PRJNA984425).

According to the methods of Caporaso et al. [16], QIIME v1.8.0 package, Lima v1.7.0 and Cutadapt 1.9.1 software were used to process the original data and cluster as operational taxonomic units (OTUs). Venn diagram and rarefaction curve were performed at OTU level. The alpha diversity difference was measured by ACE, Chao1, Shannon, Simpson indices, the level of classification was OTU, and Student *t* test was performed to analyze the significance. The beta diversity difference was performed via principal coordinate (PCoA) and nonmetric multi-dimensional scaling (NMDS) analysis, the distance algorithm selected weighted_unifrac [17]. Microbiota compositions at the phylum and species levels were measured using the naive Bayes classifier via the SILVA database and calculated using QIIME 1.7.0 [18]. Line discriminant analysis effect size (LEfSe) analysis was performed for searching the significantly different biomarkers with 3.5 LDA threshold value [19]. The predictive functional gene analysis was performed by PICRUST2 based on the KEGG database with level 2 of hierarchy [20]. These results were performed using BMKCloud (www.biocloud.net, accessed on 3 April 2023) and analyzed with the Student *t* test (*p* < 0.05) in IBM SPSS Statistics 22.0 software.

### 2.5. Integrated Analysis of Proteomics and Microbiotas

Differentially expressed proteins and microbial OTUs from the nasal mucus of pangolins were used for integrated analysis. The Spearman correlation was used for this analysis as it performs better with normalized counts (protein expression) as well as compositional data (microbiome relative abundance) compared to other metrics. The corresponding *p* values were computed by cor.test function with two-sided alternative hypothesis. These pictures were displayed using Cytoscape v3.5.1 network with appropriate threshold values (coefficient of correlation ≥ 0.6).

### 2.6. Bioinformatics Analysis of Complement C5 in Pangolins

The amino acid sequences of complement C5 from *M. javanica* (XP_036878153.1) and *M. pentadactyla* (XP_036771376.1) were compared with those from *Homo sapiens* (AAI13739.1) in the GenBank database using DNAMAN version 4.13. The SMART program (http://smart.embl-heidelberg.de/index2.cgi, accessed on 20 June 2022) was used to determine the framework regions of complement C5. The three-dimensional (3D) models of complement C5 and C5a were constructed using SWISS-MODEL program (http://swissmodel.expasy.org/interactive, accessed on 28 June 2022).

### 2.7. RNA Extraction and cDNA Synthesis of Spleen from Pangolins

The spleen samples of *M. javanica* and *M. pentadactyla* were obtained from deceased pangolins collected by our laboratory previously, with approval from the Animal Ethics Committee (Approval No. GIZ20220321). Total RNA was extracted using TRIzol Reagent (Invitrogen, Carlsbad, CA, USA) following the manufacturer’s protocols. cDNA for C5a amplification was synthesized from total RNA using a RevertAid First Strand cDNA Synthesis kit (Thermo Fisher Scientific, Waltham, MA, USA).

### 2.8. Recombinant Protein Vector Establishment and Expression

DNA fragments of C5a in *M. javanica* and *M. pentadactyla* were amplified with the primers of amplification in Appendix A. The PCR products were verified by agarose gel electrophoresis and sequencing and served as a template to amplify with primers including cleavage sites and protective bases (Appendix A) to establish the expression vector. The PCR product was digested with restriction enzymes EcoRI and HindIII (Takara, Dalian, China), ligated to the pET32a vector, and then transformed into *E. coli* BL21 (DE3) (Vazyme, Nanjing, China). The expression of C5a recombinant protein was incubated by 1 mM isopropy1-β-d-thiogalactopyranoside (IPTG, Roche, Basel, Switzerland) and examined with SDS-PAGE. 

## 3. Results

### 3.1. Protein Identification and Relative Quantification

In this study, TMT was performed to identify and detect the differentially expressed proteins in the nasal mucus between *M. javanica* and *M. pentadactyla*. The average number of proteins identified in each sample was about 510 ± 5.4 (Appendix A), suggesting a good uniformity in the type of proteins. The results of Venn diagrams were similar, as shown in Appendix A. Minor differences were observed in the protein types between the two pangolin species or the individuals.

PCA was employed to identify the relationship patterns in pangolin samples from different species and genders. Figure 1a shows a distinction in the PCA between *M. javanica* and *M. pentadactyla*, whereas minor differences were detected between the male and female pangolins. Though originating from different locations (Shenzhen and Guangzhou), the five sampling dots of *M. pentadactyla* could still be clustered together and were distant from the sampling dots of *M. javanica*, implying that environmental factors had minimal influence on proteomic results in this study, while the species differences were the main reason for expressed differences. The protein expression analysis in *M. pentadactyla* revealed 109 up-regulated proteins and 82 down-regulated proteins (Figure 1b). Volcano plots and hierarchical cluster analysis also suggested the same results for different species (Appendix A). Based on the aforementioned results, the following analyses were carried out between *M. javanica* and *M. pentadactyla* instead of male and female pangolins.

### 3.2. GO Functional Enrichment Analysis of Differentially Expressed Proteins

The abundant proteins and differentially expressed proteins were annotated and enriched using GO to further analyze the functional characteristics. As shown in Appendix A, the GO enrichment analysis of abundant proteins in pangolins showed the top 10 items of biological process, cellular component, and molecular function. There were 16 differentially expressed proteins with significant enrichment in biological processes between *M. javanica* and *M. pentadactyla*. Compared with *M. pentadactyla*, *M. javanica* possessed 12 up-regulated proteins, significantly enriched in response to endoplasmic reticulum stress (GO: 0034976), cellular response to hypoxia (GO: 0071456), positive regulation of lipid biosynthetic process (GO: 0046889), and so on (Figure 2a). Further, four down-regulated proteins were significantly enriched in iron ion transport (GO: 0006826) (Figure 2b). The information about enriched proteins related to the immune response in *M. pentadactyla* is listed in Table 1. *M. pentadactyla* had more proteins related to iron ion transport than *M. javanica*, including ceruloplasmin, ferritin, and transferrin (*p* ˂ 0.05). Abundant complement proteins, including factor B, C3, C5, and properdin, were more expressed in the nasal mucus of *M. pentadactyla*, though without significant difference (*p* = 0.052). Moreover, a similar trend was detected in the serum with statistical significance between *M. javanica* and *M. pentadactyla* (*p* ˂ 0.05, unpublished), which verified the findings in the nasal mucus.

### 3.3. Pathway Enrichment Analysis of Differentially Expressed Proteins

As described in Section 3.2, KEGG pathway enrichment analysis was performed to identify the differentially expressed proteins associated with the signaling pathways. Appendix A showed that the most enriched pathway was complement and coagulation cascades (path: cfa04610). A total of five up-regulated proteins and three down-regulated proteins in *M. javanica* (compared with *M. pentadactyla*) were significantly enriched in protein processing in endoplasmic reticulum (path: cfa04141) and inflammatory mediator regulation of TRP channels (path: cfa04750) pathways, respectively (*p* < 0.05, Figure 3). In addition, compared with *M. javanica*, *M. pentadactyla* had more proteins related to inflammatory regulation (including calmodulin, kininogen, and IL-1 receptor accessory protein) in the nasal mucus (*p* < 0.05). The findings were shown in Table 2.

### 3.4. Nasal Microbiota Identification and Diversity Analysis

Considering different proteins involved in bacterial invasion, the microbiome analysis was performed to find the discrepant germs in pangolins. We analyzed the general pattern of each sample and deleted the sample named Z–Z4 because of the huge difference compared with others. The number of OTUs in each sample is shown in Appendix A. The flat trend in the rarefaction curve indicated reasonable sampling, suggesting enhanced reliability in subsequent analysis (Figure 4b). The Venn diagram of OTUs in pangolins indicated 251 common OTUs between *M. javanica* and *M. pentadactyla*, with a large number of unique OTUs in each species (Figure 4a). 

A comparison of alpha and beta diversities between pangolins was shown in Figure 5. No significant differences were found in Chao1 and ACE indices, suggesting similar richness in *M. javanica* and *M. pentadactyla* (*p* > 0.05). The Shannon and Simpson indices in *M. javanica* were more than those in *M. pentadactyla*, representing increased community evenness in *M. javanica*, but without a significant difference (*p* > 0.05). The PCoA and NMDS analyses indicated significant differences between *M. javanica* and *M. pentadactyla* (*p* < 0.05), similar to the aforementioned results. 

### 3.5. Differential Microbiota and Predicted Function between Pangolins 

The composition analysis of nasal microbiota was detected to identify any significantly different bacteria between *M. javanica* and *M. pentadactyla*. As shown in Figure 6a, most of the bacteria identified belonged to the phyla Firmicutes and Proteobacteria (total > 85%). *M. javanica* exhibited a higher proportion of Firmicutes (74.3% > 34.2%) and a lower proportion of Proteobacteria (12.8% < 59.5%) compared with *M. pentadactyla*. At the species level, *M. javanica* showed a higher abundance of *Staphylococcus_simulans* (23.7% > 9.3%) and a lower abundance of unclassified_*Moraxella* (5.4% < 51.2%) compared with *M. pentadactyla*. The LEfSe analysis was used to find the bacteria with significant differences between pangolins (Figure 6b,c). More enrichment of *Vagococcus_lutrae*, *Nosocomiicoccus_ampullae*, unclassified_*Erysipelothrix*, and unclassified_*Helicobacter* was observed in *M. javanica* (*p* < 0.05). *M. pentadactyla* had a significantly higher abundance of *Tropheryma_whipplei* and unclassified_*Moraxella* (*p* < 0.05). 

KEGG functional forecast and difference analysis were performed using PICRUST2 analysis. As shown in Figure 7, the predicted functions related to Nucleotide metabolism and Cellular community-prokaryotes were significantly higher in *M. javanica* (*p* < 0.05). The predicted function of Amino acid metabolism in *M. pentadactyla* was significantly higher than that in *M. javanica* (*p* < 0.05).

### 3.6. Integrated Analysis of Proteomics and Microbiotas between Pangolins 

Proteomics and microbiotas conjoint analysis was implemented to explain the correlation between the discrepant proteins and bacteria in the nasal mucus of pangolins. As shown in Figure 8a, multiple differentially expressed proteins were significantly relevant to the bacteria in the nasal mucosa (*p* < 0.05), especially *Vagococcus_lutrae*, *Nosocomiicoccus_ampullae*, and unclassified_*Moraxella*, which were significant differences between pangolins. Next, we drew the heat map to detect the correlation with the genera of significantly different bacteria mentioned above and complement proteins (Figure 8b). *Vagococcus* and *Nosocomiicoccus* were significantly positive and relevant to complement factor B, C3, C7, and C9 of *M. javanica* (*p* < 0.05). Nevertheless, in *M. pentadactyla*, complement C5 presented a negative relationship with *Moraxella* (*p* < 0.05), and despite the positive correlation, it was still related to complement factors B, C3, and C4a (*p* < 0.05). In the iron transport system (Figure 8c), only the ceruloplasmin of *M. javanica* presented a negative relationship with *Moraxella* (*p* < 0.05); other proteins had no significant correlation with bacteria (*p* > 0.05). However, the enriched *Moraxella* in *M. pentadactyla* was significantly positive and relevant to ferritin, lactotransferrin, and serotransferrin (*p* < 0.05). 

### 3.7. Multiple Sequence Alignment and 3D-Modeling of Complement C5 in Pangolins 

The proteomics analysis revealed that complement C5 in the serum of *M. javanica* was significantly less than *M. pentadactyla* (*p* < 0.05, Figure 9a) hence, multiple sequence alignments were performed between pangolins. Figure 9b showed the sequences of complement C5 (partial) and C5a in *M. javanica* had obvious deficiency, whereas these in *M. pentadactyla* were unabridged. Therefore, the functional motifs of complement C5 in pangolins and humans were aligned, as shown in Figure 9c. The ANATO, the main domain of the C5a [21], was deleted in the C5 of *M. javanica* compared with *M. pentadactyla* and H. sapiens. Based on the homology modeling of human C5 (5i5k), the 3D modeling of complement C5 molecules in pangolins was analyzed (Figure 9d). C5a in *M. pentadactyla* and *H. sapiens* had 4 α-helixes forming a stalk (yellow portions in the box). The same deficiency was detected in C5a of *M. javanica*, without any conformation in the stalk. The aforementioned results might predict the functional loss of complement C5a in *M. javanica*.

### 3.8. DNA Amplification and Recombinant Protein Verification of C5a in Pangolins 

The full lengths of C5a were amplified using spleen cDNA from pangolins to verify the deficiency in *M. javanica*. As shown in Figure 10a, the length of C5a in *M. pentadactyla* was about 250 bp, which was obviously longer than that in *M. javanica* (less than 100 bp). Moreover, the recombinant protein analysis of pangolin C5a further verified the difference between *M. javanica* and *M. pentadactyla* (Figure 10b). After IPTG induction, the main strong band at ~24 kDa was observed in the expressive strain of C5a in *M. javanica*. In contrast, the molecular weight of the major band in the expressive strain from *M. pentadactyla* was about 36 kDa. The discrepancy in the C5a proteins was verified between *M. javanica* and *M. pentadactyla*, suggesting different functions.

## 4. Discussion

The mucosal immune system is found in many organs and tissues throughout the body, can activate innate and adaptive immune responses, and serves as the first-line defense against pathogenic microorganisms [22]. As a part of the mucosal immune system, the nasal mucus possesses abundant proteins involved in physiologic and pathologic processes [23]. The study on nasal mucus is an effective way to avoid causing damage to pangolins, considering their status as first-grade state-protected animals in China. This study used the TMT-based proteomics approach to investigate the differences in nasal mucus protein expression patterns between *M. javanica* and *M. pentadactyla* to further understand the differences in their immunity. More than 660 proteins were identified, and 191 proteins showed differential expression between the 2 pangolin species. The significantly up-regulated or down-regulated annotated proteins involved in critical processes of the immune response were listed in Table 1 and Table 2.

Iron plays a crucial role as a trace element in sustaining the life of organisms, primarily acquired through food intake and entering the biological body in the form of Fe^2+^. Under the influence of ceruloplasmin, free Fe^2+^ undergoes oxidation to Fe^3+^, subsequently binding to either ferritin or transferrin to mitigate the potential toxic effects of iron ions [24]. Ceruloplasmin, an α2-globulin containing copper ions expressed by liver cells, serves as a non-toxic transporter and repository of copper. Beyond its oxidase-like function, ceruloplasmin acts as a reactive protein in the acute phase of inflammation [25]. Ferritin, a complex comprising deferritin and the iron core, exhibits a robust ability to store iron, thereby maintaining the body’s iron supply and the relative stability of hemoglobin. Much like C-reactive protein, ferritin also serves as a marker for inflammation [26]. This study showed that *M. javanica* displayed lower levels of ceruloplasmin and ferritin in nasal mucus compared to *M. pentadactyla*. This discrepancy suggests potential distinctions in the two pangolins responses to early inflammation. Transferrin (including lactotransferrin, serotransferrin, etc.) plays a pivotal role not only in the transport and metabolism of iron ions but also in various physiological and biochemical functions. For instance, transferrin contributes to antibacterial effects by competitively binding iron to bacteria, which is essential for normal bacterial physiology [27]. Additionally, transferrin has been linked to ferroptosis in cells, indicating its potential value in resisting persistent infections caused by intracellular bacteria [28,29]. The observed differential expression of lactotransferrin and serotransferrin in the two pangolins may indicate variations in their antimicrobial abilities, with *M. pentadactyla* appearing to necessitate a more robust antibacterial capacity for maintaining health.

In view of the above findings, we conducted microbiome analyses on the nasal mucus of the two pangolin species to investigate whether differences in microbiota could explain different protein profiles observed. In *M. javanica*, the enriched *Erysipelothrix* spp. revealed extremely high diversities, and only a few strains owned high pathogenicity (*Erysipelothrix*_*rhusiopathiae*) [30]. *Vagococcus* spp. was used in aquaculture as a probiotic agent for *Sparus aurata* and *Dicentrarchus_labrax*, but its safety was questionable in mammals [31,32]. *Nosocomiicoccus_ampullae* was identified and reported in 2008, but reports of illnesses are still lacking [33]. The significantly enriched strains of *M. javanica* were probably common bacteria without confirmed pathogenicity. Nevertheless, we found that the enriched strains in *M. pentadactyla* were more likely to cause diseases. *Tropheryma_whipplei*, the main pathogen in Whipple’s disease, is found in humans with low antibody levels or HIV infection and causes intestinal malabsorption and systemic chronic inflammation [34,35]. *Moraxella* is a conditioned pathogen on the mucosal surface of warm-blooded animals. This study identified a new species belonging to the *Moraxella* genus in pangolins. It revealed high conservation with *Moraxella_ovis* and *Moraxella_nasovis*, which could lead to inflammation at the sites of invasion [36,37]. The conjoint analysis revealed a significantly positive correlation between *Moraxella* and ferritin as well as transferrin in *M. pentadactyla*, suggesting that ion transport proteins play a crucial role in the surveillance and inhibition of pathogenic bacteria. Therefore, we hypothesized that *M. pentadactyla* might keep health via a robust transferrin system, effectively resisting the invasion of pathogenic microorganisms.

Complement is a series of membranous and soluble immune proteins with various biological functions after activation, including eradication of circulating immune complexes, killing of pathogenic bacteria, cytolysis, opsonophagocytosis, and inflammatory disease production [38]. Complement C3 and C5 are the linchpins in complement system activation. The small fragments named C3a and C5a are lysed from them and released into the circulatory system under the effect of C3 and C5 convertase; the remainder, C3b and C5b, participate in subsequent membrane attack complex formation [39]. Complement factor B and properdin can facilitate the generation of C3 convertase and activate the complement alternative pathway, which is the first line of defense against pathogen infection [40]. In the present study, the lower expression level of the complement proteins in *M. javanica* indicated that *M. javanica* might possess lower complement activation or have functional defects in the complement system.

A mass of studies have demonstrated the relationship between mucosal bacteria and body proteins essential for normal physiological activities and immune system responses, especially in enteric microorganisms [41,42]. Complement proteins play important parts in resisting bacterial infection of the innate immune system, while bacteria have also evolved mechanisms to evade innate immunity and complement. For instance, *Streptococcus_pneumoniae* and *Neisseria_meningitidis* have been verified to inhibit activation by binding to complement inhibitors [43,44]. A similar strategy has been detected in *Moraxella*, which could restrain complement function by binding C4b-binding protein, Factor H, and C3d [45,46]. In this study, significantly enriched unclassified_*Moraxella* was negatively related to complement C5 in *M. pentadactyla*, implying that it might reduce C5 to escape complement surveillance, which agreed with *Moraxella*_*lacunata* infecting humans [47]. Additionally, unclassified_*Moraxella* had no effect on complement factors B, C3, and C4a, which performed functions normally to watch the bacterial invasion of epithelial cells.

Human C5a is a glycoprotein with a molecular weight of 11 kDa and consists of 4 nonparallel α-helixes linked with 3 disulfide bonds. As with C3a and C4a, C5a is also a kind of anaphylatoxin that mediates the release of abundant inflammatory mediators and proinflammatory cytokines by activating the mastocytes, basophils, and platelets, further leading to angiectasis, capillary permeability enhancement, and smooth muscle contraction, and ultimately mediating local inflammation [48]. C5a is known as the strongest complement molecule in inflammation mediation; it performs many biological functions, including contracting the smooth muscle and enhancing the vascular permeability, working as a chemokine to pull the neutrophils, eosinophils, and monocytes, and stimulating the mastocytes and neutrophils to produce oxygen radicals, arachidonic acid, and a large number of inflammatory factors (prostaglandins, IL-1, IL-6, and so on) [49]. The right amount of C5a is beneficial to enhancing the defense against pathogenic microorganism infection. Nevertheless, excess C5a can result in systemic inflammatory reactions and multiple organ dysfunction syndrome, causing serious damage to the body [50]. Here, we were surprised to find that the major structure of C5a was deficient in *M. javanica*, which was verified by multiple predictions and recombinant expression, causing dysfunction. These might also explain why the C5 or C5-like protein of *M. javanica* is not detected in this proteomics. After considering the aforementioned functions of C5a, we speculated that *M. pentadactyla* exhibited a higher level of inflammation compared to *M. javanica*, which was supported by the findings indicating higher levels of reactive proteins in inflammation for *M. pentadactyla*.

Similar results were corroborated in the proteomic analysis of differentially expressed proteins of proteomic analysis (Table 2). Calmodulin is an important protein related to calcium ion transduction, which widely exists in various eukaryotes. It performs vital biological functions such as regulating cellular metabolism, modulating target esterase activity, mediating muscle contraction, facilitating the inflammatory response, and so on [51,52]. Kininogen produced by hepar is the precursor of kinin [53]. Kinin can cause not only redness, swelling, hotness, and aches in different parts of the body but also vasodilation and smooth muscle contraction as a key target in inflammatory reactions [54]. Moreover, activating the kallikrein–kininogen–kinin system can indirectly activate the C3 convertase and further activate the complement [55]. Interleukin-1 receptor accessory protein (IL-1RAP), alias interleukin-1 receptor 3 (IL-1R3), forms the high-affinity IL-1R complex via associating with IL-1R1, consequently activating NF-κB with IL-1-dependent mediation of inflammation [56]. This study showed that *M. javanica* had fewer proteins mentioned above, suggesting a low level of inflammation. Furthermore, the lower complement activation in *M. javanica* was verified by the down-regulated protein expression of kininogen. These findings validated that *M. pentadactyla* exhibited more pathogens via an enhanced transferrin system and elevated levels of inflammation for maintaining health. However, excessive inflammation might also damage the body, which caused a diminished success rate in rescuing *M. pentadactyla*.

## 5. Conclusions

In summary, we detected the functional differences in nasal mucus between pangolins via TMT-based proteomics and microbiome analyses. These findings indicated that *M. pentadactyla* owned more pathogenic bacteria and neutralized through a powerful transferrin system. *M. javanica* possessed stronger anti-inflammatory ability, which might be due to the structural deficiency of C5a. Future research should focus on the application of the C5a inhibitor in *M. pentadactyla* to achieve a better survival rate.

## Figures and Tables

**Figure 1 animals-14-02683-f001:**
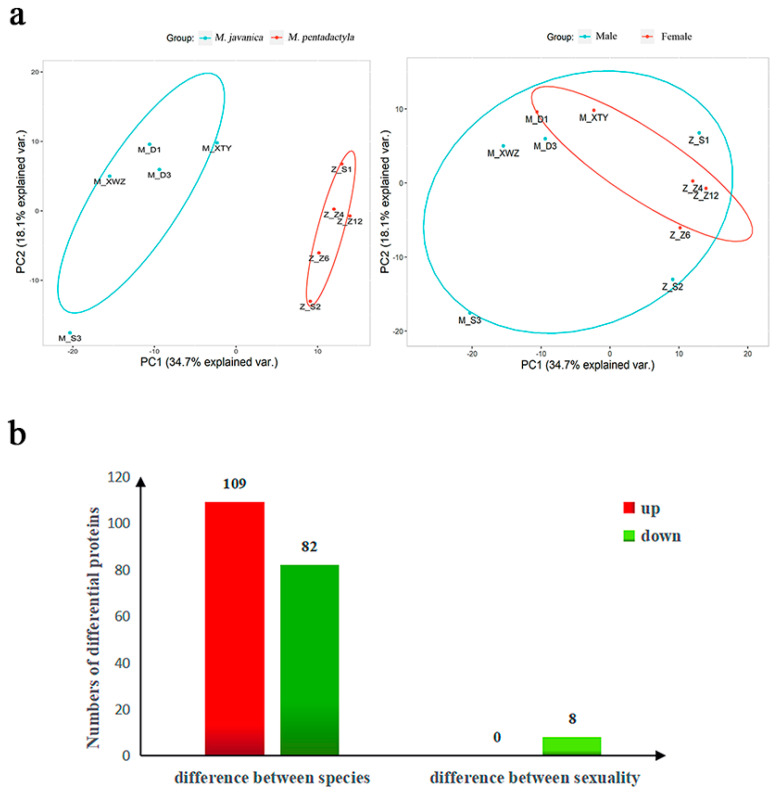
PCA and differentially expressed proteins statistics of the nasal mucus in pangolins. (**a**, **left**) PCA of *M. javanica* and *M. pentadactyla*; (**a**, **right**) PCA of male and female pangolins; the dots named M− mean *M. javanica*, and the dots named Z− mean *M. pentadactyla*, the detailed information about samples were provided in Appendix A. (**b**) Numbers of up-/down-regulated proteins in different species and sexuality (based on the expression of proteins in *M. pentadactyla* and male pangolins).

**Figure 2 animals-14-02683-f002:**
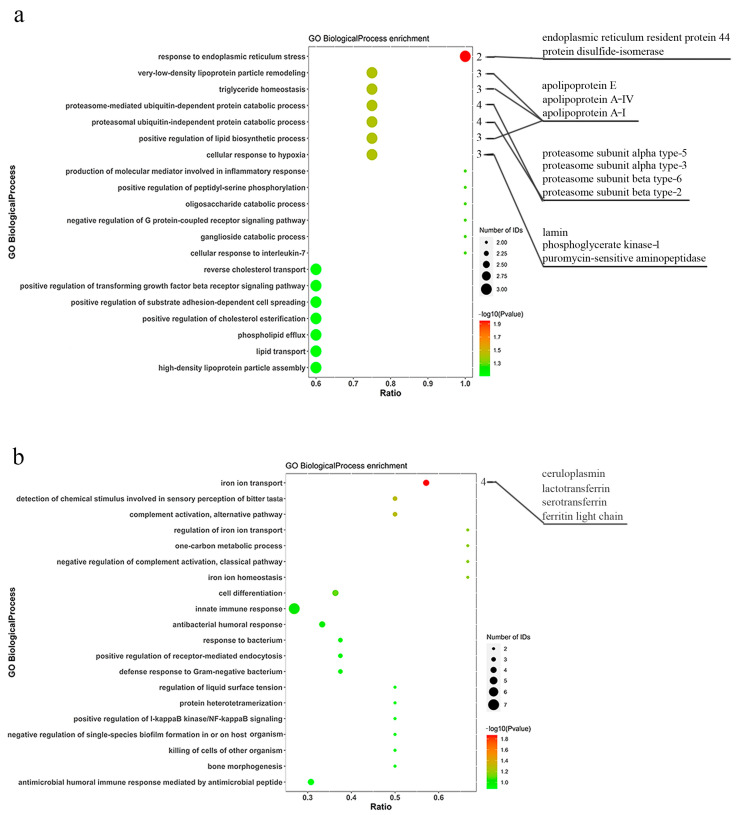
GO classification of differentially expressed proteins between *M. javanica* and *M. pentadactyla*. (**a**) Increased enrichment (*M. javanica* compared with *M. pentadactyla*); (**b**) Decreased enrichment (*M. javanica* compared with *M. pentadactyla*). The numbers indicated the number of proteins significantly enriched in each term. The statistical significances were achieved using *t* test, with *p* < 0.05, Fold change > 1.2.

**Figure 3 animals-14-02683-f003:**
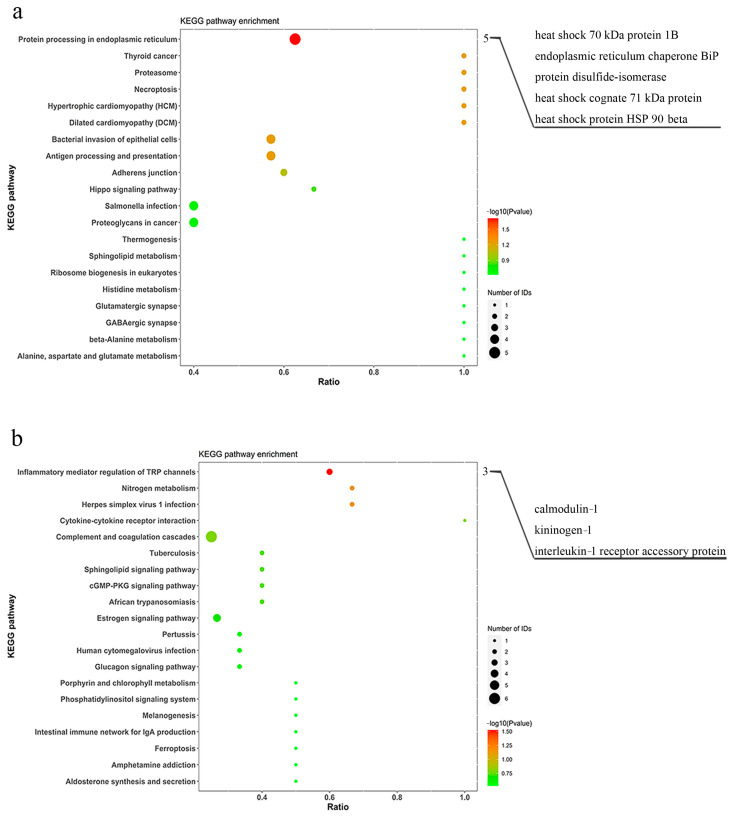
KEGG classification of differentially expressed proteins between *M. javanica* and *M. pentadactyla.* (**a**) Increased enrichment (*M. javanica* compared with *M. pentadactyla*); (**b**) Decreased enrichment (*M. javanica* compared with *M. pentadactyla*). The numbers indicated the number of proteins significantly enriched in each term. The statistical significances were achieved using *t* test, with *p* < 0.05, Fold change > 1.2.

**Figure 4 animals-14-02683-f004:**
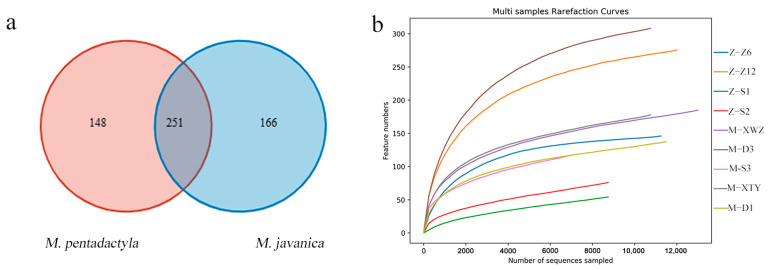
Venn diagram of OTUs (**a**) and rarefaction curve (**b**) of the nasal microbiota in *M. javanica* and *M. pentadactyla*.

**Figure 5 animals-14-02683-f005:**
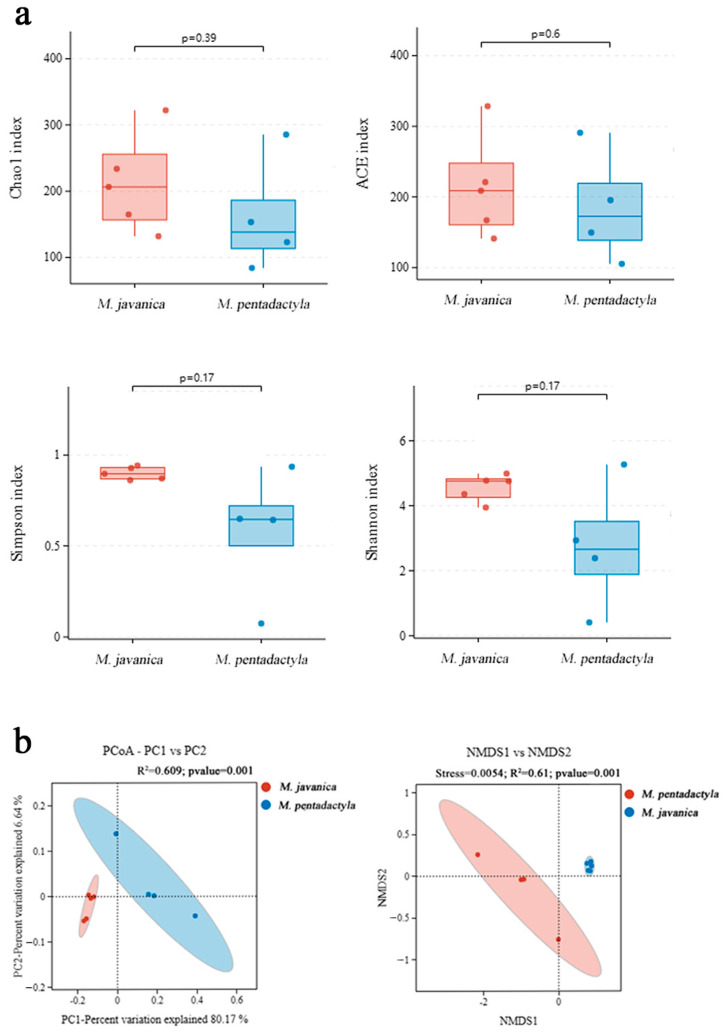
Alpha (**a**) and beta (**b**) diversity analyses of nasal microbiota between *M. javanica* and *M. pentadactyla*.

**Figure 6 animals-14-02683-f006:**
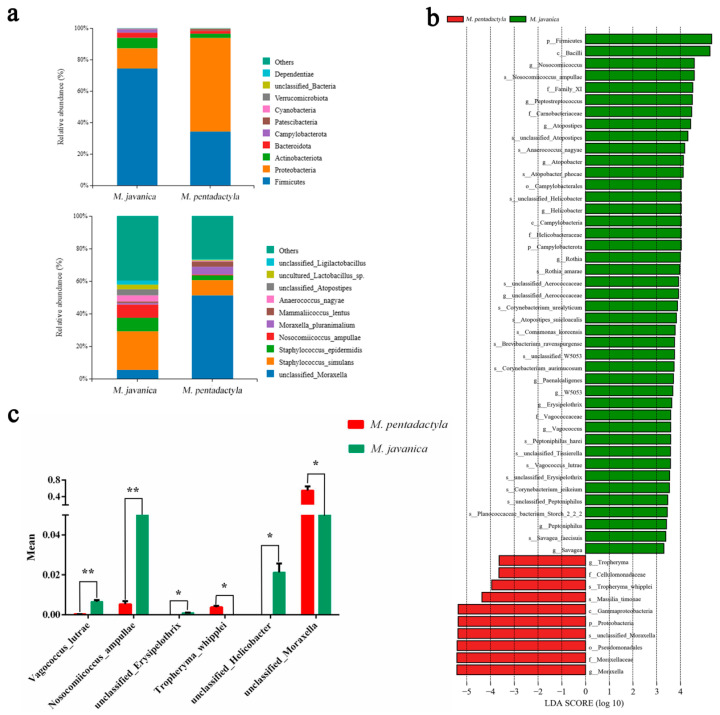
Composition and difference analysis of nasal microbiota between *M. javanica* and *M. pentadactyla*. (**a**) Composition of nasal microbiota at the phylum (**up**) and species (**down**) levels; (**b**) LEfSe analysis of bacteria with significantly different abundance in *M. javanica* and *M. pentadactyla*; (**c**) Significantly different bacteria in the nasal mucus between *M. javanica* and *M. pentadactyla*. Bars represented the results from four/five pangolins. Significant differences in the abundance of bacteria were indicated with an asterisk (*p* < 0.05) or two asterisks (*p* < 0.01).

**Figure 7 animals-14-02683-f007:**
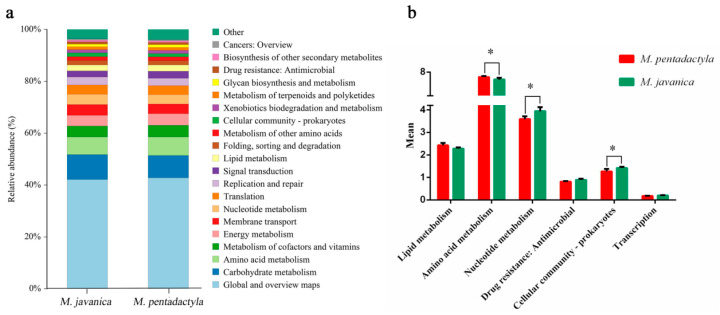
KEGG function forecast and difference analysis of nasal microbiota between *M. javanica* and *M. pentadactyla*. (**a**) KEGG function forecast of nasal microbiota at the species level with level two functional hierarchy; (**b**) Significantly different predicted functions between *M. javanica* and *M. pentadactyla*. Bars represented the results from four/five pangolins. Significant differences in functions were indicated with an asterisk (*p* < 0.05).

**Figure 8 animals-14-02683-f008:**
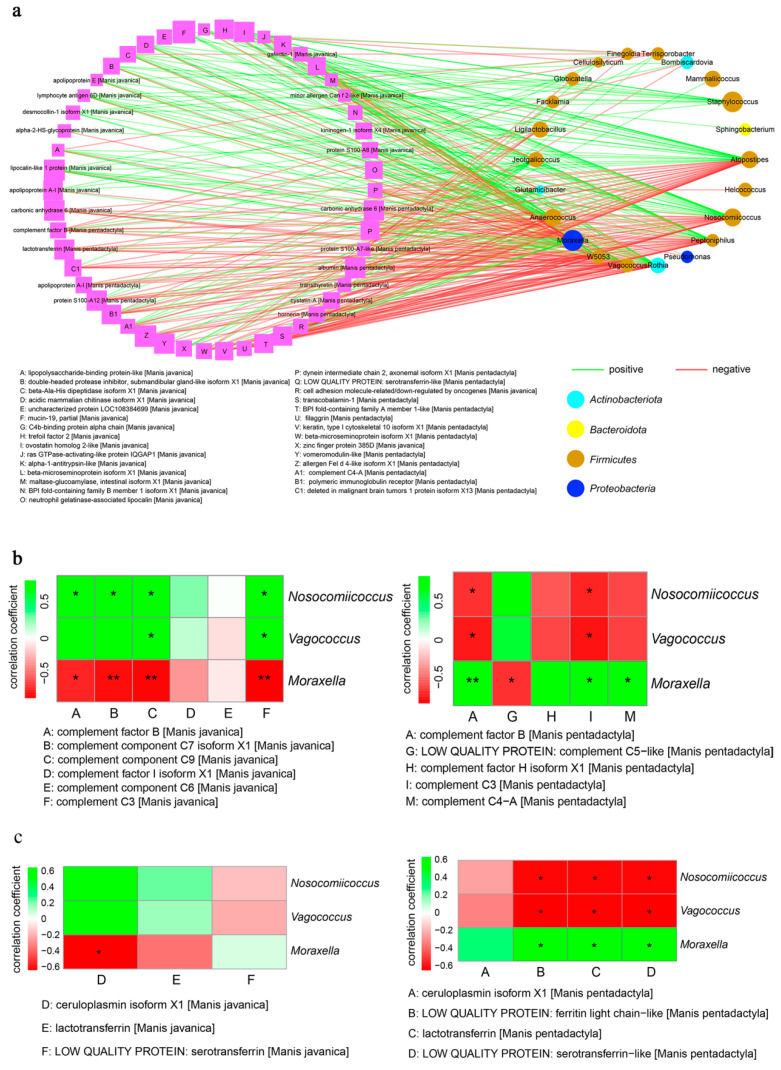
Proteomics and microbiotas conjoint analysis of nasal mucus in *M. javanica* and *M. pentadactyla*. (**a**) The correlation between differentially expressed proteins and discrepant bacteria. Squares represented proteins of pangolins, circles represented bacteria in the nasal mucosa, and the volumes represented the abundance. The green lines represented positive correlation, red lines represented negative correlation, and the line thickness represented the levels of correlation. Coefficient of correlation, ≥0.6; *p*-value ≤ 0.05. (**b**) The correlation between significantly discrepant bacteria and complement proteins in *M. javanica* (**left**) and *M. pentadactyla* (**right**), respectively. The green boxes represented positive correlation; red boxes represented negative correlation. Significant differences were indicated with an asterisk (*p* < 0.05) or two asterisks (*p* < 0.01). (**c**) The correlation between significantly discrepant bacteria and iron transport proteins in *M. javanica* (**left**) and *M. pentadactyla* (**right**), respectively. The green boxes represented positive correlation; red boxes represented negative correlation. Significant differences were indicated with an asterisk (*p* < 0.05).

**Figure 9 animals-14-02683-f009:**
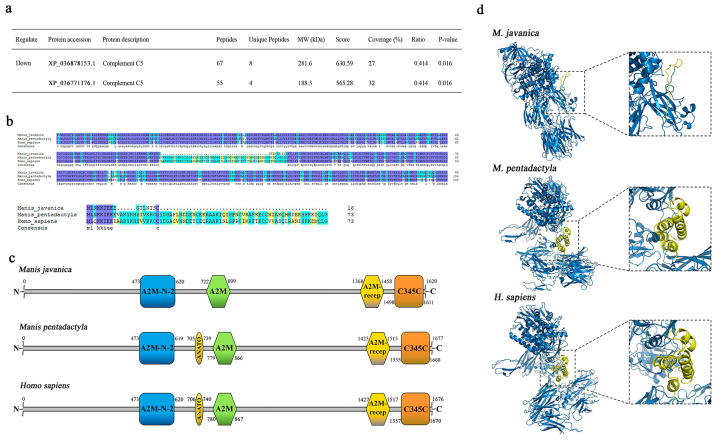
Bioinformatics analysis of complement C5 in *M. javanica*, *M. pentadactyla*, and *H. sapiens*. (**a**) Serum complement C5 of *M. javanica* expressed significantly lower versus *M. pentadactyla* (*p* < 0.05); (**b**) Protein sequence alignment of complement C5 (partial) and C5a; (**c**) Comparison of functional motifs of complement C5. The functional motifs were indicated in the named color boxes. (**d**) Comparison of 3D models of complement C5. The yellow sections in the box showed C5a in pangolins and humans. The parameters in the 3D model are: (*M. javanica*) GMQE: 0.74, QMEAN: 0.76; (*M. pentadactyla*) GMQE: 0.7, QMEAN: 0.71; (*H. sapiens*) GMQE: 0.69, QMEAN: 0.71.

**Figure 10 animals-14-02683-f010:**
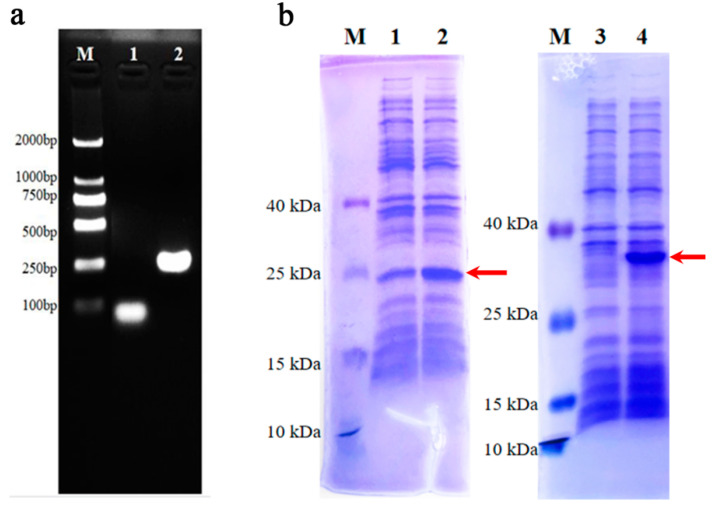
Gene amplification and prokaryotic recombinant protein expression of *M. javanica* and *M. pentadactyla*. (**a**) Amplification of C5a full-length from the cDNA of pangolins. M: marker, 1: C5a of *M. javanica*, C5a of *M. pentadactyla*; (**b**) SDS-PAGE analysis of C5a connected with a carrier. M: marker, 1: recombinant protein of C5a from *M. javanica* without induction; 2: recombinant protein of C5a from *M. javanica* with induction; 3: recombinant protein of C5a from *M. pentadactyla* without induction; 4: recombinant protein of C5a from *M. pentadactyla* with induction. The red arrows represent the C5a recombinant proteins of *M. javanica* and *M. pentadactyla*, respectively.

**Table 1 animals-14-02683-t001:** Enriched proteins of *M. pentadactyla* compared to *M. javanica* associated with immune response based on GO analysis.

Protein Accession	Protein Description	Peptides	Unique Peptides	MW (kDa)	Score	Coverage (%)	Ratio	*p*-Value
XP_036752253.1	ceruloplasmin	17	2	124.9	54.99	19	0.57	0.011
XP_036739216.1	lactotransferrin	63	12	76.8	552.35	66	0.57	0.011
XP_036778791.1	ferritin	2	2	34.9	1.96	5	0.57	0.011
XP_036738762.1	serotransferrin	52	12	110.7	298.14	40	0.57	0.011
XP_036731391.1	complement factor B	20	2	88.1	87.86	27	0.5	0.052
XP_036746741.1	complement C3	57	10	186.1	184.87	34	0.5	0.052
XP_036771376.1	complement C5-like	2	1	188.4	4.31	1	0.5	0.052
XP_036750838.1	properdin	2	2	50.8	7.06	5	0.5	0.052

**Table 2 animals-14-02683-t002:** Enriched proteins of *M. pentadactyla* compared to *M. javanica* associated with immune response based on KEGG analysis.

Protein Accession	Protein Description	Peptides	Unique Peptides	MW (kDa)	Score	Coverage (%)	Ratio	*p*-Value
XP_036748218.1	calmodulin-3-like	3	3	16.5	16.06	36	0.6	0.030
XP_036730991.1	kininogen-1	12	3	47.7	33.9	28	0.6	0.030
XP_036731091.1	IL-1 receptor accessory protein	2	2	78.1	5.15	2	0.6	0.030

## Data Availability

The original data of microbiome have been uploaded to SRA data in National Center for Biotechnology Information (NCBI) database (PRJNA984425). The original data of proteomics have not been uploaded in the interest of maintaining confidentiality. Nevertheless, we are prepared to provide Appendix A should the editors or reviewers deem it necessary.

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
