# Peer review of "Proteomics and Microbiota Conjoint Analysis in the Nasal Mucus: Revelation of Differences in Immunological Function in Manis javanica and Manis pentadactyla"

_animals, 2024, doi:10.3390/ani14182683_

Round 1

Reviewer 1 Report

Comments and Suggestions for Authors

The following are the concerns,

1.      In the methods section, the authors should specify clearly the version and the date of access of the reference database in the UniProt.  Authors should explain, how the matching was made, when the samples were of two different pangolin species. Was crossmatching performed or different species?

2.      What is meant by Perseus algorithm to compensate the missing data? The authors should be detailed, how the compensation was made, either lowest value or using statical methods.

3.      The authors should clearly state the parameters including the number of unique peptides of the protein identification.

4.      In the results section, the authors should specify one number rather than a range, “proteins identified and quantified was about 506–513”

5.      The authors should also deposit the proteome data in repository such as ProteomeXchange.

Comments on the Quality of English Language

Quality is good

Author Response

Comments 1: In the methods section, the authors should specify clearly the version and the date of access of the reference database in the UniProt.  Authors should explain, how the matching was made, when the samples were of two different pangolin species. Was crossmatching performed or different species?

Response 1: Thanks for your valuable suggestions. The data interpretation and protein identification were carried out using the latest Manis protein databases (downloaded from NCBI, 4. 3. 2023 version), configured with Proteome Discoverer 2.4 software for searching the datasets. The peptide sequences were searched in the database of M. javanica and M. pentadactyla, and a protein must have at least one unique peptide match with TMT ratios. We have rewrited the  methods section of how raw MS data was searched in the database, the information above in the revised manuscript (Line 107-111 and 117-119).

Comments 2: What is meant by Perseus algorithm to compensate the missing data? The authors should be detailed, how the compensation was made, either lowest value or using statical methods.

Response 2: Perseus algorithm is a conventional source code via stochastic compensation. Hence it is neither lowest nor statical value. We have supplemented the information in the revised manuscript (Line 119-121).  

Comments 3: The authors should clearly state the parameters including the number of unique peptides of the protein identification.

Response 3: Thanks for your helpful suggestions. The data interpretation and protein identification were carried out using the latest Manis protein databases (downloaded from NCBI, 4. 3. 2023 version), configured with Proteome Discoverer 2.4 software for searching the datasets. The search parameters include 15 ppm and 0.02 Da mass tolerances for MS and MS/MS, respectively. Trypsin digestion allowing two missed cleavage with fixed modifications of cysteine carbamidomethylation, N-terminal TMT10plex, and lysine TMT10plex; and with Acetyl and oxidation (M) as variable modifications. The value of max missed cleavages was 2. Additionally, the peptides were extracted using high peptide confidence. The calculation of a 1% false discovery rate (FDR) was used for searching the peptide sequence. The peptide sequences were searched in the database of M. javanica and M. pentadactyla, and a protein must have at least one unique peptide match with TMT ratios. We have rewrited the methods section 2.3, and supplemented the parameters about protein database search in the revised manuscript (Line 108-119).

Comments 4: In the results section, the authors should specify one number rather than a range, “proteins identified and quantified was about 506–513”

Response 4: According to your suggestions, we have rewrited the sentence in the revised manuscript (Line 192-194). 

Comments 5: The authors should also deposit the proteome data in repository such as ProteomeXchange.

Response 5: According to your suggestions, the raw MS data were available via ProteomeXchange with identifier PXD054168. We have supplemented the information in the revised manuscript (Line 131).

Reviewer 2 Report

Comments and Suggestions for Authors

An integrated proteomic and micobiome analysis of nasal mucus of two species of critically endangered pangolins (Manis) was performed to clarify the differences in the immune level responses between M. javanica and M. pentadactyla. The data covering both proteome and microbiome on nasal mucus in pangolin is unique and has value in its own remit as a rich dataset.  Subsequent informatics prediction, recombinant protein verification, and nasal mucosa microbiome association analysis were performed integrally to deepen the understanding of the functional differences between pangolins.

However, I have concerns about the data presentations and therefore the following details must be provided.

Authors have to provide detailed descriptions of how raw MS data was searched against Uniprot database of two species, esp, search setting mass tolerance, modifications included, FDR control, filters applied based on numbers of peptides if any,  and how multiple protein accession numbers  (of both species) were handled in DB search.

 Authors are required to submit raw MS data including metadata to proteomics repositories (eg. PRIDE/Massive/ProteomeXchange etc).

Its not clear how they have counted the protein numbers in each sample (Supplementary Fig S1 and S2, protein statistics and Venn diagrams). Given they have followed TMT labelling, which is an isobaric labelling technique and therefore cannot be traced back to individual samples upon mixing, except at the quantification based on reported ion area. Authors need to elaborate how protein numbers were derived from each sample, once isobaric labelling is performed.

 State clearly what data has been used to create PCA plots, (Reporter Ion intensities?) in Fig 1 Legends.

Line 202 states, ‘screened abundant proteins’, but not clear what the authors refer to. If the authors mean differentially expressed proteins, then re-phrase the sentence.  

Source of the data presented in Fig 9a (plasma Compliment C5) is not stated and how was this performed?

Comments on the Quality of English Language

 I haven't noticed any major issues around the English language and seems appropriate.

Author Response

Comments 1: Authors have to provide detailed descriptions of how raw MS data was searched against Uniprot database of two species, esp, search setting mass tolerance, modifications included, FDR control, filters applied based on numbers of peptides if any,  and how multiple protein accession numbers  (of both species) were handled in DB search.

Response 1: Thanks for your valuable suggestions.The data interpretation and protein identification were carried out using the latest Manis protein databases (downloaded from NCBI, 4. 3. 2023 version), configured with Proteome Discoverer 2.4 software for searching the datasets. The search parameters include 15 ppm and 0.02 Da mass tolerances for MS and MS/MS, respectively. Trypsin digestion allowing two missed cleavage with fixed modifications of cysteine carbamidomethylation, N-terminal TMT10plex, and lysine TMT10plex; and with Acetyl and oxidation (M) as variable modifications. The value of max missed cleavages was 2. Additionally, the peptides were extracted using high peptide confidence. The calculation of a 1% false discovery rate (FDR) was used for searching the peptide sequence. The peptide sequences were searched in the database of M. javanica and M. pentadactyla, and a protein must have at least one unique peptide match with TMT ratios. We have supplemented the information above in the revised manuscript (Line 108-119).

Comments 2: Authors are required to submit raw MS data including metadata to proteomics repositories (eg. PRIDE/Massive/ProteomeXchange etc).

Response 2: According to your suggestions, the raw MS data were available via ProteomeXchange with identifier PXD054168. We have supplemented the information in the revised manuscript (Line 131).

Comments 3: Its not clear how they have counted the protein numbers in each sample (Supplementary Fig S1 and S2, protein statistics and Venn diagrams). Given they have followed TMT labelling, which is an isobaric labelling technique and therefore cannot be traced back to individual samples upon mixing, except at the quantification based on reported ion area. Authors need to elaborate how protein numbers were derived from each sample, once isobaric labelling is performed.

Response 3: The number of proteins in each sample is counted by the number of proteins that have a quantitative value. In TMT labeling experiment, although the samples are mixed together, each sample carries a different label, so that statistics can be based on the number of proteins in each sample.

Comments 4: State clearly what data has been used to create PCA plots, (Reporter Ion intensities?) in Fig 1 Legends.

Response 4: Principal Component Analysis (PCA) is a statistical technique that synthesizes a new set of uncorrelated variables from the original proteins, based on their expression patterns. This method aims to capture the essence of the original variables while effectively reducing their dimensionality. The PCA diagram were produced based on the composition and content of protein in each sample. Moreover, we have supplemented the information of each dots in Fig 1 legends (Line 213-215).

Comments 5: Line 202 states, ‘screened abundant proteins’, but not clear what the authors refer to. If the authors mean differentially expressed proteins, then re-phrase the sentence.  

Response 5: Sorry for your misunderstanding. We have rewrited the sentence in the revised manuscript (Line 218-222) for enhanced accuracy.

Comments 6: Source of the data presented in Fig 9a (plasma Compliment C5) is not stated and how was this performed?

Response 6: The figure 9a was from our another paper about the differential expression analysis in serum proteomic between M. javanica and M. pentadactyla (unpublished). A similar trend of expression difference in complement proteins was detected in the serum with statistical significance (P = 0.016), which was verified the findings in the nasal mucus. So we speculated the differences of complement proteins in the nasal mucus were significative, though without significant difference (P = 0.052).

Round 2

Reviewer 1 Report

Comments and Suggestions for Authors

The authors have revised the manuscript and the raw data is also deposited at the ProteomeXchange database

Reviewer 2 Report

Comments and Suggestions for Authors

The revised manuscript provides sufficient details about the methods and clarification on some of the results presented. When a ProteomXchange submission is cited in a manuscript for peer reviewing, the usual practice is to provide a reviewer access code, which I don't see anywhere.

Comments on the Quality of English Language

Minor language edits may be required